# *BcABF1* Plays a Role in the Feedback Regulation of Abscisic Acid Signaling via the Direct Activation of *BcPYL4* Expression in Pakchoi

**DOI:** 10.3390/ijms25073877

**Published:** 2024-03-30

**Authors:** Xiaoxue Yang, Meiyun Wang, Qian Zhou, Xinfeng Xu, Ying Li, Xilin Hou, Dong Xiao, Tongkun Liu

**Affiliations:** 1Key Laboratory of Biology and Genetic Improvement of Horticultural Crops (East China), State Key Laboratory of Crop Genetics & Germplasm Enhancement, Ministry of Agriculture and Rural Affairs of China, Engineering Research Center of Germplasm Enhancement and Utilization of Horticultural Crops, Ministry of Education of China, Nanjing Agricultural University, Nanjing 210095, China; 2021104060@stu.njau.edu.cn (X.Y.); 2021104059@stu.njau.edu.cn (M.W.); 2020204026@stu.njau.edu.cn (Q.Z.); 2023204038@stu.njau.edu.cn (X.X.); yingli@njau.edu.cn (Y.L.); hxl@njau.edu.cn (X.H.); 2Sanya Research Institute, Nanjing Agricultural University, Nanjing 210095, China

**Keywords:** *BcABF1*, *BcPYL4*, positive feedback regulation, leaf water loss, ABA signaling

## Abstract

Abscisic acid-responsive element-binding factor 1 (*ABF1*), a key transcription factor in the ABA signal transduction process, regulates the expression of downstream ABA-responsive genes and is involved in modulating plant responses to abiotic stress and developmental processes. However, there is currently limited research on the feedback regulation of *ABF1* in ABA signaling. This study delves into the function of *BcABF1* in Pakchoi. We observed a marked increase in *BcABF1* expression in leaves upon ABA induction. The overexpression of *BcABF1* not only spurred *Arabidopsis* growth but also augmented the levels of endogenous IAA. Furthermore, *BcABF1* overexpression in *Arabidopsis* significantly decreased leaf water loss and enhanced the expression of genes associated with drought tolerance in the ABA pathway. Intriguingly, we found that BcABF1 can directly activate *BcPYL4* expression, a critical receptor in the ABA pathway. Similar to *BcABF1*, the overexpression of *BcPYL4* in *Arabidopsis* also reduces leaf water loss and promotes the expression of drought and other ABA-responsive genes. Finally, our findings suggested a novel feedback regulation mechanism within the ABA signaling pathway, wherein BcABF1 positively amplifies the ABA signal by directly binding to and activating the *BcPYL4* promoter.

## 1. Introduction

Abscisic acid (ABA) is a plant hormone that plays a crucial role in resisting abiotic stress and is therefore also referred to as a ‘stress hormone’. ABA mediates plant responses to environmental stresses such as drought, cold, osmotic stress, and salt stress [1,2,3,4]. When plants are exposed to environmental stress, ABA rapidly accumulates, enhancing the plant’s adaptability to stressful conditions [5]. ABA also participates in regulating various plant developmental processes, including aging, seed germination, and root elongation [6,7,8]. The core components of the ABA signaling pathway include the abscisic acid receptors PYR/PYL/RCAR family; 2C-type protein phosphatase (PP2C) subfamily; and the SnRK2 kinase subfamily [9,10,11]. Under normal growth conditions, PP2C binds to SnRK2, inhibiting SnRK2 kinase activity [12]. When plants are under stress, ABA binds to the PYR/PYL/RCAR receptors [13]. The ABA-PYR/PYL/RCAR complex binds to PP2C, inhibiting PP2C phosphatase activity and resulting in the phosphorylation of SnRK2 [14]. The phosphorylated SnRK2 phosphorylates ABRE/ABF, activating ABF proteins. Activated ABF proteins bind to the ABRE sequence in the promoter of responsive genes, thereby activating the ABA signaling pathway and enhancing the plant’s tolerance to environmental stress [15,16]. ABA is crucial for plant growth and development and is involved in responding to environmental stress, particularly drought stress. ABA enhances the plant’s ability to clear reactive oxygen species (ROS), reducing oxidative damage and increasing drought tolerance [17,18]. For example, *MADS23*, by promoting the transcription of ABA biosynthetic genes (*NCEDs*), increases endogenous ABA levels, enhancing drought tolerance in rice [19]. Under drought conditions, ABA mediates stomal closure, reducing the rate of water loss in plants, which is an effective method to improve plant drought resistance [20]. In tomatoes, microRNA160 participates in ABA-mediated stomal regulation by inhibiting *ARF10*, reducing leaf water loss, maintaining leaf water balance, and enhancing tomato adaptability to drought stress [21].

In *Arabidopsis*, there are nine homologs of *ABF/AREB* [22]. Via functional gain and loss studies, *ABF/AREB* (*ABF1*, *AREB1*/*ABF2*, *ABF3*, and *AREB2/ABF4*) have been identified as critical transcription factors within the ABA signaling pathway. Exhibiting functional redundancy, they collaboratively regulate stomatal opening and are involved in mediating the plant’s drought response mechanisms. Relevant studies have shown that *ABRE/ABF* (*ABF1-4*) effectively enhances the expression of *ADF5* (*Arabidopsis actin depolymerizing factor 5*), which contributes to the remodeling of the actin cytoskeleton, thereby promoting the closure of stoma and subsequently enhancing the drought tolerance of plants [23]. At the same time, the interaction between ABF (ABF1-4) protein and IDD14 (INDETERMINATE DOMAIN 14) enhances ABF’s transcriptional activity. This process leads to the induction of stomatal closure and stimulates the expression of genes responsive to abscisic acid (ABA), thereby significantly enhancing drought tolerance in *Arabidopsis*. This mechanism underscores the critical role of ABA in plant adaptive responses to water stress conditions [22]. The overexpression of *ABF3* and *ABF4* leads to stomal closure and reduced transpiration [24]. In contrast, the *abf3* and *abf2 abf3 abf4* multiple mutants exhibit increased water loss rates under drought conditions [25,26]. Additionally, ABF2 and ABF4 interact with ANAC096, participating in the ABA-mediated process of stomal closure and leaf water loss [27]. In Pakchoi, four *Arabidopsis* ABF orthologs, BcABF1-4, have been identified [28]. Research on ABFs within Pakchoi remains sparse. Within the Brassicaceae family, BnaABF2 engages in stoma closure via its interaction with the DELLA protein BnaRGA, playing a role in enhancing the drought tolerance of *Brassica napus* [29]. The overexpression of *BnaABF2* heightens the stoma’s responsiveness to ABA, thereby diminishing leaf water loss [30]. Additionally, BnaABF3 and BnaABF4 are subject to phosphorylation by BnaCPK5, which positively influences the drought resistance of *Arabidopsis* [31]. The molecular mechanisms by which ABF1 controls water loss in Pakchoi leaves are largely unexplored.

In order to prevent excessive responses to ABA signaling, plants have evolved a feedback regulation mechanism for ABA signaling. This mechanism not only maintains the homeostasis of ABA signaling but also promotes adaptive responses to environmental changes. Current research has demonstrated that *ABF/AREB* is involved in the feedback regulation of ABA signal transduction and ABA biosynthesis processes. *ABFs* negatively regulate the ABA signal via feedback by promoting the expression of A-group *PP2C* genes [32]. *FYVE1*, an intermediate regulatory factor in the ABA signaling pathway’s negative feedback regulation process, binds to *PYR/PYL/RCAR* receptors and promotes their degradation [33,34,35]. FYVE1 can also be phosphorylated by SnRK2, and phosphorylated FYVE1 inhibits the transcriptional activation activity of *ABF4*. *ABF4*, by directly binding to the *FYVE1* promoter, regulates *FYVE1* expression, participating in the negative feedback regulation of the ABA signaling pathway [36]. The bZIP transcription factor ABI5 regulates seed germination via feedback regulation of the expression of *PYL11* and *PYL12* [37]. Therefore, the in-depth study of feedback regulation in the ABA pathway is crucial for understanding its significant role in plant growth and development. Clade E Growth-Regulating 1 and 2 (*EGR1/2*) positively regulate plant ABA signaling by modulating the phosphorylation of SnRK2.2 [38]. *PYLs* have been identified as crucial elements in the positive feedback modulation of ABA signaling. Via the interaction and phosphorylation of MdPYL2/12, MdCPK4 plays a significant role in enhancing ABA signal transduction [39].

Pakchoi [*Brassica campestris* (syn. *Brassica rapa*) ssp. *chinensis*] features a brief growth cycle, with the vegetative phase spanning 1–2 months. This vegetable is favored for its ease of cultivation, high productivity, superior traits, and nutritional richness, establishing it as a vegetable of worldwide acclaim. Nevertheless, in the cultivation process, Pakchoi is vulnerable to abiotic stresses, including salt stress [40], cold stress [41], and heat stress [42]. Its relatively shallow and dispersed root system, while beneficial for efficiently absorbing surface soil nutrients and moisture, renders it particularly susceptible to drought conditions. Consequently, drought poses a significant constraint on the growth of Pakchoi. Our research findings indicate that in Pakchoi, *BcABF1* can enhance the plant’s drought resistance by reducing leaf water loss. By binding directly to the promoter of *BcPYL4*, BcABF1 regulates the ABA signaling pathway via a positive feedback loop, thereby enhancing ABA signaling and rapidly increasing the drought tolerance of Pakchoi.

## 2. Result

### 2.1. Phylogenetic Analysis of ABF1

To elucidate the evolutionary relationship of ABF1, we selected seven cruciferous plant species (*B. napus*, *B. campestris*, *B. carinata*, *B. nigra*, *B. juncea*, *B. oleracea*, and *Arabidopsis thaliana*) for analysis. We used DNAMAN for amino acid sequence alignment and the SMART online analysis tool for the prediction of functional structures. Specifically, the amino acid sequence of ABF1 in *B. campestris* was found to be 82.68% similar to *B. oleracea*, 96.73% to *B. napus*, 96.46% to both *B. nigra* and *B. juncea*, 89.3% to *B. carinata*, and 73.53% to *Arabidopsis thaliana* (Figure 1a). The results indicated that all the amino acid sequences exhibited high identity and contained the conserved BRLZ domain (Figure 1a). To further understand the evolutionary relationship of ABF1, we constructed a phylogenetic tree based on the ABF1 amino acid sequences of these species (Figure 1b). The branching characteristics revealed that BcABF1 and BnABF1 are on the same evolutionary branch, indicating a close genetic relationship between *B. campestris* and *B. napus*.

### 2.2. Expression Analysis of BcABF1

To obtain the tissue-specific expression of *BcABF1* in Pakchoi, qRT-PCR analysis was completed on the root, stem, leaf, stalk, petiole, and flower. Our findings revealed that the expression level of *BcABF1* was the highest expression in the leaves, with the second highest expression observed in the flowers (Figure 2a). To advance our understanding of the *BcABF1* expression pattern, we engineered a GUS reporter system under the control of the *BcABF1* promoter (*pBcABF1:GUS*) and transformed it into *Arabidopsis thaliana* (Col-0), resulting in the creation of the *pBcABF1:GUS/Col*. GUS staining of this line revealed a notable staining intensity in the leaves, primarily concentrated in the vascular and stoma (Figure 2b). Previous research has documented diurnal rhythm variations in *ABF* [43]. We conducted a qRT-PCR analysis to assess the diurnal variations in ABF1 expression over a 24 h period in 14 d Pakchoi seedlings (Figure 2c). This analysis revealed that the expression of *BcABF1* was significantly higher during daylight hours compared to nighttime. Notably, the expression peaked at Zeitgeber Time (ZT) 8. Following this peak, there was a gradual decrease in expression. Subsequently, to explore ABF1’s responsiveness to ABA, we applied 50 μM ABA to 14-day Pakchoi seedlings at ZT0. Our findings revealed that, relative to untreated Pakchoi, *BcABF1*’s expression levels surged within the initial 8 h window post-treatment, diminished noticeably from 12 to 24 h, and eventually stabilized at an elevated level by the 24 h mark. Specifically, for post-ABA treatment, the most substantial shift in *BcABF1* expression occurred at ZT8, where it was threefold higher than that in untreated plants (Figure 2c).

These results indicate that *BcABF1* is primarily expressed in leaves and is ABA-inducible, displaying a strict diurnal rhythm change.

### 2.3. Overexpression of BcABF1 in Arabidopsis Reduces Leaf Water Loss

To study the function of *BcABF1*, we engineered the *35S:BcABF1* expression vector and successfully transformed it into *Arabidopsis*, leading to the creation of stable 3*5S:BcABF1* lines (#1, #2) in Col-0 (Figure 3a and Appendix A). Following qRT-PCR analysis, the expression level of BcABF1 was found to be higher in line #2. Hence, line #2 was selected. Intriguingly, we noticed an increase in the overall leaf size of the *35S:BcABF1* (Figure 3a). To understand the factors driving this morphological alteration in *35S:BcABF1*, we measured the dimensions of the largest leaf in 28-day *35S:BcABF1* and WT. While the WT had a leaf length of 51.40 mm and a width of 21.75 mm, the *35S:BcABF1* exhibited significantly larger leaves, with lengths and widths measuring 63.85 mm and 24.60 mm, respectively (Figure 3b,c). This clear disparity in leaf size, with 3*5S:BcABF1* leaves being notably longer and wider than those of the WT. To verify this finding, we compared the fresh and dry weights of the *35S:BcABF1* and WT. The analysis revealed that *35S:BcABF1* displayed an increase in both fresh and dry weights in contrast to WT (Appendix A). These findings indicate that the heterologous expression of *BcABF1* in *Arabidopsis* promotes the growth of plants. Studies have demonstrated the critical role of auxins in the growth and development of plants [44]. Among them, Indole-3-acetic acid (IAA) stands out not only as the principal auxin but also as one of the auxins with the highest physiological activity [45]. Further, utilizing the ELISA technique, we quantified the IAA content in 28-day *35S:BcABF1* and WT. The IAA concentration in WT was found to be 617 ng/g, whereas a marked increase to 648 ng/g was observed in *35S:BcABF1* (Appendix A). These findings imply a possible connection between *BcABF1* and auxin levels.

Studies have demonstrated that *ABRE/ABF* signaling can upregulate the expression of *ADF5*, which contributes to the remodeling of the actin cytoskeleton, thereby promoting the closure of stoma and subsequently enhancing the drought tolerance of plants [23]. In an effort to investigate the effects of *BcABF1* expression in stomata on the drought resistance capabilities of plants, we conducted an assessment of the leaf water loss rate in 21-day *Arabidopsis.* This involved specifically analyzing the aerial parts of the plants. Notably, when compared to the WT, the *35S:BcABF1* (#2) variant exhibited a substantially lower rate of leaf water loss at the 60 and 80 min marks during the dehydration process, highlighting its enhanced drought resilience. As expected, the detached leaves of *35S:BcABF1* exhibited significantly reduced water loss compared to WT (Figure 3d). To delve deeper into the impact of *BcABF1* on enhancing plant drought resistance, specifically via the reduction in leaf water loss, we concentrated our efforts on assessing the expression of ABA-responsive genes that are known to be associated with drought resistance. These genes include *RD29A*, *RD29B*, *RAB18*, and *KIN2* [46]. Our investigation involved comparing these genes in both WT and *35S:BcABF1*. We observed that the expression levels of *RD29A*, *RD29B*, *RAB18*, and *KIN2* in the *35S:BcABF1* were significantly elevated in comparison to the WT (Figure 3e–h). These results further indicate that *BcABF1* can improve plant drought resistance by reducing leaf water loss.

### 2.4. BcABF1 Directly Binds to the BcPYL4 Promoter

*ABFs*, key transcription factors in the ABA signaling pathway, are located downstream and primarily regulate the expression of ABA-responsive genes. Interestingly, in *35S:BcABF1*, the expression of *BcPYL4* was increased (Figure 4a). *PYL4*, known as a crucial member of the PYR/PYL receptor family, is located upstream of *ABF* and plays a vital role in maintaining the normal operation of the ABA signaling pathway [47,48]. We assumed that *ABF1* might be involved in feedback regulating the expression of *PYL4*, thereby modulating the ABA signaling pathway. To test this hypothesis, we analyzed the *BcPYL4* 2000 bp promoter and found the presence of an ABRE element (−916) (Appendix A). Next, we used the Yeast One-Hybrid Assay (Y1H) to check whether BcABF1 binds to the *BcPYL4* promoter region. The assay showed that BcABF1 can directly bind to the *BcPYL4* promoter (Figure 4b). Furthermore, we conducted an analysis on the promoter cis elements of additional PYR/PYL/RCAR receptor family members. Our findings revealed the presence of ABRE binding sites on the promoters of *BcPYR1*, *BcPYL2*, *BcPYL3*, *BcPYL5*, *BcPYL7*, *BcPYL8*, and *BcPYL11* (Appendix A). However, using Y1H analysis to further study the interaction between BcABF1 and these promoters, we discovered that BcABF1 can only bind to the promoter of *BcPYL2* (Appendix A).

Subsequently, to delve deeper into *BcABF1′s* regulatory role on *BcPYL4*, a dual-luciferase assay was executed. Notably, the LUC/REN ratio in plants co-transformed with *35S:BcABF1-GFP* and *pBcPYL4-3-0800* demonstrated a significant increase compared to the control (Figure 4c,d). This suggests that BcABF1 actively binds to the ABRE motifs in the *BcPYL4* promoter, thereby enhancing its expression (Figure 4c,d). Subsequently, our focus shifted to assessing whether BcABF1 could activate *BcPYL4* within a living organism. For this purpose, a *GUS* (β-glucuronidase) reporter gene, driven by the *BcPYL4* promoter (*pBcPYL4:GUS*), was constructed and introduced into Col-0 and *35S:BcABF1 Arabidopsis*. In line with our predictions, the *pBcPYL4:GUS* signal was observable in the stoma of *pBcPYL4:GUS/Col*. More strikingly, GUS staining was markedly more intense in the *35S:BcABF1/pBcPYL4:GUS* line compared to the control *pBcPYL4:GUS/Col* (Figure 4e). Furthermore, there was a significant increase in GUS activity in the *35S:BcABF1/pBcPYL4:GUS* (Figure 4f). These findings lead us to conclude that BcABF1 is capable of directly binding to the *BcPYL4* promoter, thereby activating its transcription.

### 2.5. Overexpression of BcPYL4 Amplifies the ABA Signaling Pathway

To further investigate the role of *BcPYL4* in the ABA signaling pathway, a stable *Arabidopsis* line expressing *35S:BcPYL4* was established (#7, #8, and #12) (Figure 5a,b). We also conducted leaf water loss experiments on three lines of *35S:BcPYL4* (Figure 5c). The results indicated that there was a reduction in the rate of leaf water loss in these three lines compared to WT, especially at 60 min and 100 min. This is similar to the dehydration phenotype of our *BcABF1* overexpressing *Arabidopsis*. To advance our exploration of *BcPYL4′s* role in enhancing drought resistance, we conducted an analysis focusing on the expression of drought response genes *RD29A* and *RAB18* (Figure 5d,e). In the *35S:BcPYL4* strains, there was a significant increase in the expression levels of *RD29A* and *RAB18*, reaching ten times that of the WT. The gene expression pattern observed in *35S:BcPYL4* closely parallels that seen in *35S:BcABF1*, indicating a uniform enhancement of drought resistance mechanisms facilitated by *BcPYL4*.

Moreover, the ABA signaling pathway not only regulates plant drought resistance but also plays a role in leaf aging, flowering, and seed germination processes [49]. Subsequent analysis of the expression changes in leaf aging genes (*NYC1* and *NYE1*), flowering-related gene (*SOC1*), and seed germination-related gene (*CYS5*) in *35S:BcPYL4* showed an upregulation of these response genes (Figure 5f–i). These indicate that overexpression of *BcPYL4* amplifies the ABA signaling pathway, not only enhancing plant drought resistance but also participating in the regulation of ABA-induced leaf aging, flowering, and seed germination processes.

## 3. Discussion

ABF1, a member of the bZIP family, exhibits a high degree of conservation across plant species and features the BRLZ domain [50]. In our study, we explored ABF1 proteins across seven cruciferous plant species. Our findings revealed that ABF1 shares similar structural features, characterized by a BRLZ domain present in all ABF1 proteins. Notably, a close evolutionary relationship between *B. campestris* and *B. napus* was observed (Figure 1), highlighting evolutionary similarities in ABF1 functions. In *Arabidopsis*, the expression of all *ABFs* is induced by ABA, drought, and low-temperature conditions [51]. Remarkably, *BcABF1* is also ABA-induced and primarily expressed in stomata and vascular tissues (Figure 2). Furthermore, ABFs exhibit diverse functions in evolution; studies suggest that *ABF2*, *ABF3*, and *ABF4* play pivotal roles in regulating chlorophyll degradation and leaf senescence processes [52]. In this study, we identified the positive impact of *BcABF1* on reducing leaf water loss and orchestrating the expression of ABA-responsive genes during this process (Figure 3). The mechanism by which *BcABF1* reduces leaf water loss remains to be fully understood. It has been reported that *ABFs* play a positive role in the regulation of stomatal closure. The interaction between ABF1 and IDD14 is implicated in the modulation of stomatal closure, resulting in a further decrease in leaf surface water loss and, to some extent, enhancing drought resistance in Arabidopsis [22]. Overexpression of *ABF3* and *ABF4* induces partial stomatal closure, leading to a reduction in water loss [24]. ANAC096, interacting with ABF2 and ABF4, actively participates in the regulation of ABA-mediated stomatal closure and water loss, further enhancing plant drought resistance [27]. Therefore, based on our data, we have reason to believe that the reduction in leaf water loss by *BcABF1* is mediated via the regulation of stomatal closure. However, the specific regulatory mechanism still requires further exploration.

Additionally, our data indicate that the overexpression of *BcABF1* in *Arabidopsis* leads to an enlarged phenotype (Figure 3a–c). These findings are intriguing because, in recent research, the overexpression of bZIP transcription factors often results in impaired plant growth. For instance, overexpression of *Osbzip48* significantly reduces plant height, ascribed to the direct binding of Osbzip48 to *Osko2* and the subsequent regulation of its expression, causing dwarfism in rice (*Oryza sativa* L.) [53]. Similarly, overexpression of *OsABF1* in rice also exhibits a semi-dwarf phenotype [54]. Another study reports that transgenic *Arabidopsis* expressing *TaABF3* shows a slight reduction in the size of rosette leaves during growth [50]. It is worth noting that genetic functional diversity arising from species genomic differences can result in phenotypic variations [55]. In our data, the overexpression of *BcABF1* in *Arabidopsis* displays an enlarged phenotype (Figure 3a–c). In this regard, we explored potential causes. According to pertinent studies, auxin plays a crucial role in plant growth and development [44]. Indole-3-acetic acid (IAA) serves not only as a major auxin but is also one of the most physiologically active auxins [45]. Therefore, we measured the IAA content in *35S:BcABF1* (Appendix A). Our results show a significant increase in IAA content in *35S:BcABF1*, providing an explanation for the observed enlarged phenotype. Studies have indicated that ABFs interact with IDD14, regulating the ABA signaling pathway [22]. Simultaneously, IDD14 directly activates *TRYPTOPHAN AMINOTRANSFERASE OF ARABIDOPSIS 1*, *PIN-FORMED1*, and *YUCCA5*, thereby regulating auxin synthesis and transport [56]. Moreover, ABA modulates the auxin signaling pathway by regulating auxin response factors *ARF5*, *ARF6*, and *ARF10* [57]. The wide-ranging interplay and crosstalk between ABA and IAA in the regulation of stress responses are well-established [58]. Under stressful conditions, increased ABA levels suppress the expression of genes encoding proteins involved in auxin transport. Conversely, under non-stressful conditions, low ABA levels stimulate auxin transport, initiating the auxin signaling pathway [59]. During normal growth, low ABA levels facilitate auxin transport, aiding in the accumulation of IAA in leaves and promoting leaf expansion in *35S:BcABF1*. Hence, we have reason to believe that *BcABF1* serves as a key junction connecting ABA and IAA hormonal signals, modulating critical nodes in plant growth and development.

The transcription factor ABF plays a pivotal regulatory role in the ABA signaling pathway, and extensive research has been carried out on ABF at both the transcriptional and translational levels [12,27,60,61]. However, there is a limited body of research exploring how ABF contributes to feedback regulation in the ABA signal. Currently, molecular mechanisms of ABA signaling have been extensively investigated. The fundamental components of the ABA signaling pathway encompass ABA receptors, PP2Cs, and SnRK2s [12,62]. In the context of feedback regulation in the ABA signal transduction pathway, ABF is recognized as a crucial regulatory factor. *PP2C*, acting as a negative regulator of ABA signaling, induces the expression of *ABFs*, leading to their dephosphorylation and inactivation, thus maintaining the equilibrium of the ABA signal [32]. ABF4 directly activates *FYVE1*, and FYVE1 binds and degrades PYL, accomplishing negative feedback regulation of the ABA signal [36]. Moreover, FYVE1 can be phosphorylated by SnRK2, and the phosphorylated FYVE1 inhibits the transcriptional activation activity of ABF4 [36]. Our data provide novel insights into the feedback regulation of ABA. BcABF1, by directly binding to the *BcPYL4* promoter, enhances *BcPYL4* expression, thus increasing ABA receptor levels (Figure 4). Overexpressing *BcPYL4* in *Arabidopsis* reduces leaf water loss and upregulates ABA-responsive gene expression (Figure 5). Consequently, we propose that BcABF1, via *BcPYL4* activation, participates in the positive feedback loop of ABA signaling. While there is functional redundancy among the 14 *PYLs* [37], not all are involved in ABF-mediated ABA feedback regulation; in Pakchoi, only *BcPYL4* and *BcPYL2* play a role (Figure 4 and Appendix A). Therefore, our data unveil an alternative regulatory pathway for *BcABF1* in ABA signaling. In this context, we establish a regulatory mechanism in plants to swiftly adapt to environmental stress by amplifying the ABA signal. *BcABF1* serves as a positive feedback regulator in the ABA signaling pathway, enhancing the ABA signal by increasing ABA receptor levels. Simultaneously, to prevent excessive response or ABA accumulation, ABF functions as a negative feedback regulator. It achieves this by inducing *PP2C* expression and degrading PYL, strictly maintaining ABA signal homeostasis.

## 4. Materials and Methods

### 4.1. Plant Materials and Growth Conditions

The Pakchoi used in this study were sourced from the Cabbage Systems Biology Laboratory at the College of Horticulture, Nanjing Agricultural University. Pakchoi, *Arabidopsis thaliana* of the ‘Columbia-0’ ecotype (Col-0), transgenic *Arabidopsis*, and *N. benthamiana* were all grown under conditions of 23 °C and a long-day photoperiod (16 h of light/8 h of dark). Col-0 and Pakchoi served as controls in our experiments, except where it was noted otherwise.

### 4.2. Arabidopsis Transformation

Using Pakchoi cDNA as a template, the coding sequences (CDS) of *BcABF1* and *BcPYL4* were amplified. The *BcABF1* CDS was then cloned into the pMDC43 vector to obtain the *35S:BcABF1* expression vector. The *BcPYL4* CDS was inserted into the pRI101 vector to obtain the *35S:PYL4* expression vector. Sequencing confirmed the correctness of both expression vectors.

The *35S:BcABF1* and *35S:BcPYL4* expression vectors were separately introduced into *Agrobacterium* (GV3101) and transformed into Col-0 using the floral dip method [63]. T_3_ generation homozygous overexpressing *Arabidopsis* lines were obtained via screening. The primers used for amplifying sequences are listed in Appendix A.

### 4.3. β-Glucuronidase Staining and Expression

Using Pakchoi genomic DNA as a template, a 2000 bp fragment of the upstream 5′ region of the *BcPYL4* gene was amplified. This promoter fragment was then cloned into the PBI121 expression vector to generate the *BcPYL4:GUS* expression vector (primers listed in Appendix A). After sequencing confirmation, the vector was introduced into *Agrobacterium* (GV3101) and separately transformed into T_3_ generation homozygous overexpressing *35S:BcABF1* lines and Col-0 using the floral dip method [63]. Selection using kanamycin on the growth medium resulted in the acquisition of homozygous T_3_ lines for *pBcPYL4:GUS/Col* and *pBcPYL4:GUS/35S:BcABF1*. As previously mentioned, the GUS enzyme activity was measured [64].

### 4.4. Quantitative Real-Time PCR (qRT-PCR)

Total RNA from plant samples was extracted using the SteadyPure Plant RNA Extraction Kit (Accurate Biotechnology (Hunan) Co., Ltd., Changsha, China) and cDNA was synthesized using the *Evo M-MLV* Mix Kit with gDNA Clean for qPT-PCR (Accurate Biotechnology (Hunan) Co., Ltd., China). Quantitative real-time PCR (qRT-PCR) was performed on the CFX96 system using qRT-PCR Master Mix (Without ROX) (Vazyme Biotech Co., Ltd., Nanjing, China). Quantitative results were calculated using the 2^−ΔΔCT^ method. Normalization in *Arabidopsis* and Pakchoi was conducted using *ELF4A* (AT1G80000) and *BcPP2A* (BraC07g034860.1), respectively [65,66]. The primers used for qRT-PCR are listed in Appendix A.

### 4.5. Yeast One-Hybrid Assay (Y1H)

To investigate the interaction between *BcABF1* and the *BcPYL4* promoter, the Matchmaker™ Gold Yeast One-Hybrid System from Clontech was employed for a yeast one-hybrid assay (Y1H). For this purpose, *BcABF1* was integrated into the pGADT7 vector, while the 2 kb promoter region of *BcPYL4* was cloned into the pAbAi vector. After screening for suitable AbA concentrations, the interaction between *pAbAi-BcPYL4* and BcABF1 was validated using the lithium acetate method (Clontech, Mountain View, CA, USA, Cat.630439).

Genes homologous to the *Arabidopsis PYR/PYLs* family were extracted from the Pakchoi genome and assessed for their interaction with *BcABF1* at the *BcPYLs* promoters using the Yeast One-Hybrid (Y1H) assay. The methodology employed for this verification was consistent with the previously described approach.

### 4.6. Dual-Luciferase Reporter Assay

The *BcABF1* gene sequence was inserted into the pRI101 vector, with *35S:BcABF1-GFP* and *35S:GFP* serving as effector vectors. The *BcPYL4* promoter was inserted into the pGreenII-0800-LUC vector as a reporter vector (primers listed in Appendix A). Following a 2:1 ratio of reporter vector to effector vector, the mixture was infiltrated into tobacco leaves. After 72 h, D-luciferin was injected into the tobacco leaves, followed by a 5 min dark incubation. Fluorescence intensity was then measured using a live plant imaging system (LUC). The LUC/REN ratio reflects binding activity.

### 4.7. Bioinformatics Analysis

The *BcABF1* sequence information was obtained from the Non-heading Chinese Cabbage and Watercress Database (http://tbir.njau.edu.cn/NhCCDbHubs/index.jsp, accessed on 1 March 2022), while homologous *ABF1* sequences from six Brassicaceae plants were acquired from BRAD (http://brassicadb.cn/, accessed on 1 March 2022). Further, we employed DNAMAN9 software (version no.10.0.2.100) to perform a multiple sequence alignment of ABF1 amino acids across seven cruciferous species. Additionally, the conserved domains of these proteins were analyzed using the SMART online software (http://smart.embl.de/, accessed on 1 March 2023) platform, and an evolutionary tree was constructed using MEGA X (version no.10.1.7, neighbor-joining algorithm, bootstrap replications = 1000).

### 4.8. Determination of IAA Content

An amount of 0.1 g of fresh *Arabidopsis* leaves from 28-day WT and *35S:BcABF1* were ground thoroughly with liquid nitrogen, and 900 μL of PBS buffer was added and mixed thoroughly. It was centrifuged at 4 °C and 8000 rpm for 30 min, and the supernatant was used for further analysis. The IAA content is measured using the Auxin ELISA Kit (BYabscience (Nanjing) Co., Ltd., Nanjing, China) according to the instructions. In the enzyme-linked immunosorbent assay (ELISA) approach, we blend a specific quantity of solid-phase antibodies with biotin-tagged IAA (indole-3-acetic acid), Auxin and non-labeled antigens (either calibration standards or test samples), to initiate a competitive inhibition reaction. Once the reaction achieves equilibrium, a complex of solid-phase antibodies linked to biotinylated IAA, Auxin is formed. Subsequently, enzyme-tagged avidin is introduced, resulting in the formation of a complex involving solid-phase antibody-biotinylated IAA, Auxin bound to enzyme-labeled avidin. Following the color development of the substrate, the absorbance (OD value) is measured at a 450 nm wavelength using an ELISA reader, providing valuable insights into the assay’s outcomes.

### 4.9. Leaf Water Loss

To accurately assess the rate of leaf water loss, we chose aerial sections from *35S:BcABF1*, *35S:BcPYL4*, and WT, ensuring they were of comparable size and at a uniform developmental stage. Each sample was composed of the aerial parts from 10 individual plants. These samples were cultivated under conditions mirroring those of plant growth. Subsequently, each sample was weighed at specific intervals (0, 20, 40, 60, 80, 100, and 120 min). Leaf water loss was quantified by the reduction in fresh weight, and the leaf water loss rate was calculated using the formula: (fresh weight − dry weight)/fresh weight, expressed as a percentage. This method provides a precise measure of water loss in leaves under varying conditions.

### 4.10. Statistical Analysis

All experiments were conducted independently at least three times, with statistical analysis of the data performed using Student’s *t*-test. *p* < 0.05 or *p* < 0.01.

## Figures and Tables

**Figure 1 ijms-25-03877-f001:**
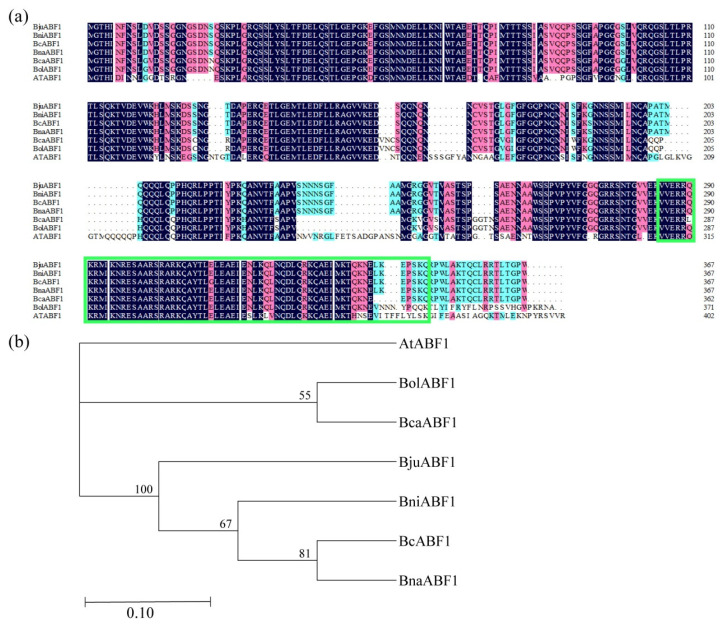
Phylogenetic analysis of ABF1. (**a**) Amino acid sequence alignment of ABF1 in seven cruciferous plant species. Identical and similar amino acids are indicated with black and pink shading, respectively, while highly variable amino acids are represented with blue shading. The part marked by the green rectangle is the BRLZ domain. (**b**) Phylogenetic tree. Phylogenetic tree was constructed using MEGA X (neighbor-joining algorithm, bootstrap replications = 1000). The numbers beside the branches represent bootstrap values from 1000 replicates. Relative changes along the branches are indicated by the scale bar. Bc*: Brassica campestris.* Bna*: Brassica napus.* Bju*: Brassica juncea.* Bni*: Brassica nigra.* Bol*: Brassica oleracea.* Bca*: Brassica carinata.* At*: Arabidopsis thaliana*.

**Figure 2 ijms-25-03877-f002:**
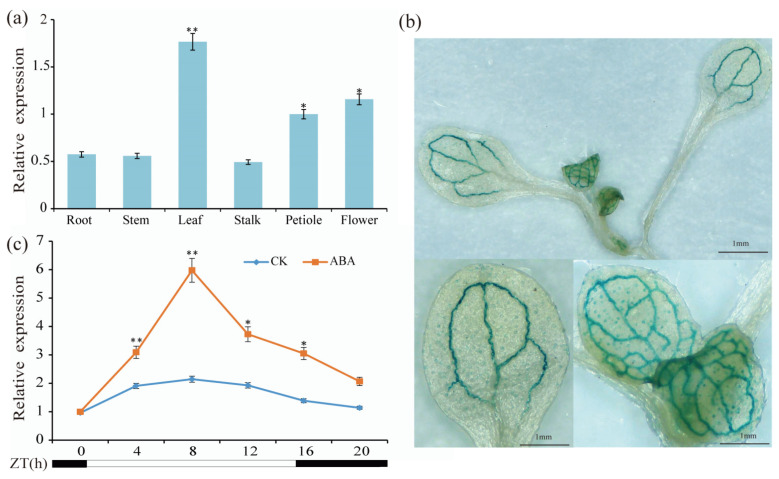
*BcABF1* expression analysis. (**a**) Expression levels of *BcABF1* in different tissues of Pakchoi. (**b**) GUS staining of *pBcABF1:GUS/Col*. (**c**) Pakchoi seedlings were treated with 50 μM ABA under long-day conditions (LD), with spraying twice a week for two consecutive weeks. Samples were collected every 4 h starting from ZT0. qPT-PCR analysis was conducted to measure the expression levels of *BcABF1*. The black areas represent dark conditions, while the white areas indicate light conditions. The data represent the average of three biological replicates (** *p* < 0.01, * *p* < 0.05, Student’s *t*-test).

**Figure 3 ijms-25-03877-f003:**
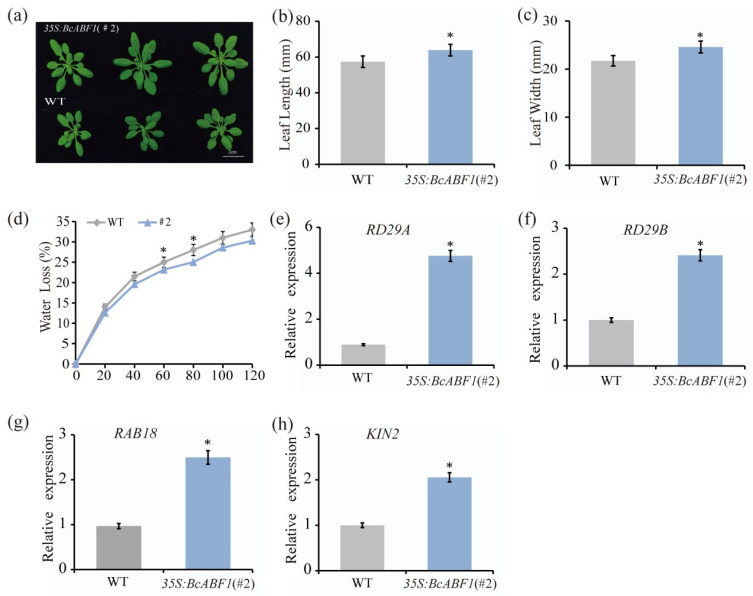
Phenotypic indicators analysis of *BcABF1* overexpressing *Arabidopsis.* (**a**) The phenotype of *Arabidopsis* overexpressing *35S:BcABF1*. (**b**) Leaf length in *35S:BcABF1*. The largest leaf from each individual 28-day 3*5S:BcABF1* and WT was selected and identified, and then its length was accurately measured. (**c**) Leaf width in *35S:BcABF1*. The largest leaf from each individual 28-day 3*5S:BcABF1* and WT was selected and identified and then its width was accurately measured. (**d**) Rate of water loss in leaves of *35S:BcABF1*. Leaves from 21-day-old *35S:BcABF1* were selected, and measurements were taken every 20 min. (**e**–**h**) Relative expression levels of ABA signaling response genes in *35S:BcABF1*. The data represent the average of three biological replicates (* *p* < 0.05, Student’s *t*-test).

**Figure 4 ijms-25-03877-f004:**
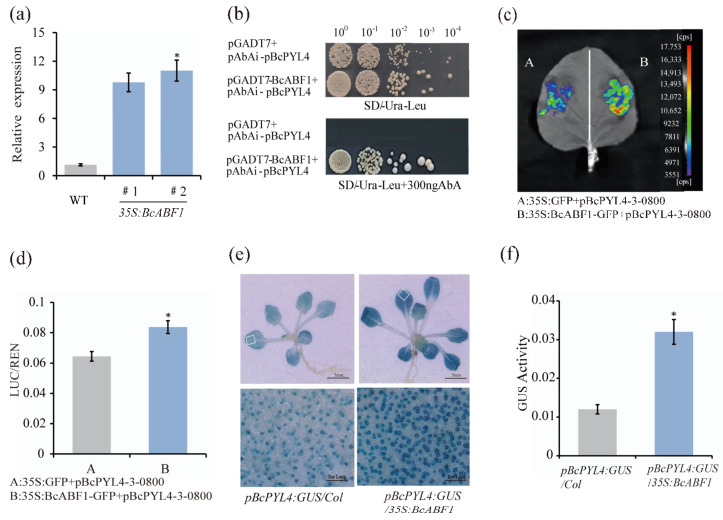
BcABF1 directly activates *BcPYL4* expression. (**a**) Relative expression level of *BcPYL4* in *35S:BcABF1*. (**b**) Y1H analysis showing direct binding of BcABF1 protein to *BcPYL4* promoter. (**c**) Imaging of LUC activity showed that BcABF1 activates the expression of *BcPYL4*. (**d**) The ratio of LUC/REN of (**c**). (**e**) GUS staining of 17-day-old *pBcPYL4:GUS/Col* and *pBcPYL4:GUS/35S:BcABF1* lines *in* which GUS gene expression was driven by the *BcPYL4* promoter. (**f**) Expression of *GUS* in *pBcPYL4:GUS/Col* and *pBcPYL4:GUS/35S:BcABF1* lines from (**e**). Data represent the average of three biological replicates (* *p* < 0.05, Student’s *t*-test).

**Figure 5 ijms-25-03877-f005:**
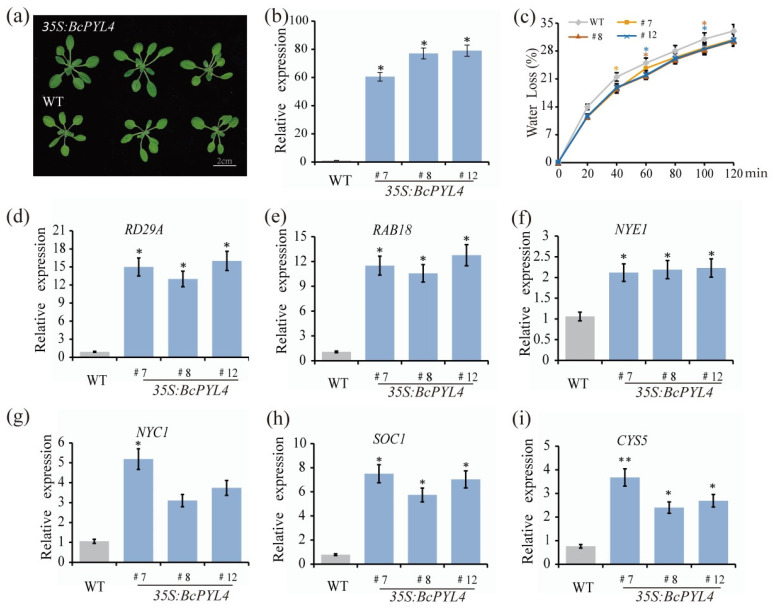
*BcABF1* positively regulates the ABA signaling pathway. (**a**) The phenotype of *Arabidopsis* overexpressing *35S:BcPYL4.* (**b**) The mRNA abundance of *BcABF1* in *35S:BcPYL4.* (**c**) Rate of water loss in leaves of *35S:BcPYL4*. Selecting similarly sized aboveground parts of 21-day-old *35S:BcPYL4*, ensuring the accurate measurement of their weight at consistent intervals of every 20 min. (**d**,**e**) Drought response genes (*RD29A*, *RAB18*) relative expression levels of in *35S:BcPYL4*. (**f**,**g**) Leaf aging genes (*NYE1*, *NYC1*) relative expression levels of in *35S:BcPYL4*. (**h**) Flowering-related gene (*SOC1*) relative expression levels of in *35S:BcPYL4.* (**i**) Seed germination-related gene (*CYS5*) relative expression levels of in *35S:BcPYL4*. Data represent the average of three biological replicates (** *p* < 0.01, * *p* < 0.05, Student’s *t*-test).

## Data Availability

Data sharing is not applicable to this article.

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
