# Peer review of "BcABF1 Plays a Role in the Feedback Regulation of Abscisic Acid Signaling via the Direct Activation of BcPYL4 Expression in Pakchoi"

_ijms, 2024, doi:10.3390/ijms25073877_

Round 1

Reviewer 1 Report

Comments and Suggestions for Authors

Dear editor.

In this manuscript titled, ‘BcABF1 Plays a Role in Feedback Regulation of ABA Signaling 2 through Direct Activation of BcPYL4 Expression in Pakchoi’. The authors have put forward a comprehensive and well written paper focusing on the expression nature and the regulatory pathway of ABA signaling in Pakchoi. This work seems valuable and, in my opinion, can be deemed to be accepted in the international journal of molecular science.

However, some minor questions and edits need to be answered as listed below.

Line 85-94. Extra information can be given on the Pakchoi, and the previous studies done on the plant.

Line 112- Indicate in the figure caption the meaning of, (a) the colors in the alignment analysis, (b) the meaning of the numbers in the phylogenetic tree, and also meaning of the bar scale.

Line 142- Arabidopsis being a model plant can produce more transgenic lines, why choose single line for analysis to work with? More lines will give more confidence on the function of the BcARF1.

Figure 4- In figure c, the legends are not clear.

Line 308- ‘in current research”? does the authors mean in recent research?

Line 393- Agrobacterium must be written in italic. Check on the rest of the manuscript.

Line 405-413- qPCR or qRT-PCR choose one and maintain consistency throughout the paper.

Line444- the meaning of ‘nine times the volume’ is not clear.

Please add the statistical analysis used in the materials and methods.

Comments on the Quality of English Language

Dear editor.

In this manuscript titled, ‘BcABF1 Plays a Role in Feedback Regulation of ABA Signaling 2 through Direct Activation of BcPYL4 Expression in Pakchoi’. The authors have put forward a comprehensive and well written paper focusing on the expression nature and the regulatory pathway of ABA signaling in Pakchoi. This work seems valuable and, in my opinion, can be deemed to be accepted in the international journal of molecular science.

However, some minor questions and edits need to be answered as listed below.

Line 85-94. Extra information can be given on the Pakchoi, and the previous studies done on the plant.

Line 112- Indicate in the figure caption the meaning of, (a) the colors in the alignment analysis, (b) the meaning of the numbers in the phylogenetic tree, and also meaning of the bar scale.

Line 142- Arabidopsis being a model plant can produce more transgenic lines, why choose single line for analysis to work with? More lines will give more confidence on the function of the BcARF1.

Figure 4- In figure c, the legends are not clear.

Line 308- ‘in current research”? does the authors mean in recent research?

Line 393- Agrobacterium must be written in italic. Check on the rest of the manuscript.

Line 405-413- qPCR or qRT-PCR choose one and maintain consistency throughout the paper.

Line444- the meaning of ‘nine times the volume’ is not clear.

Please add the statistical analysis used in the materials and methods.

Author Response

请参阅附件

Reviewer 2 Report

Comments and Suggestions for Authors

This manuscript reveals that BcABF1 plays a crucial role in the positive feedback regulation of ABA signaling. Specifically, BcABF1, a key signal of ABA, is expressed in the stomal, and its overexpression results in decreased water loss from leaves as well as an increased expression of genes related to drought resistance.  Furthermore, by utilizing Yeast One-Hybrid (Y1H) assays, dual luciferase assays, and GUS staining, this study confirms that BcABF1 contributes to the positive feedback of ABA signaling by directly activating the expression of BcPYL4.

These are my main comments:

1. The introduction sections one and two redundantly describe the importance of ABA and should be consolidated into a single paragraph.

2. The introduction should include the research progress on ABF1 in the Brassicaceae family.

3. Please ensure that the names of plant species throughout the manuscript are italicized, especially in the Materials and Methods section.

4. The manuscript exhibits issues with imprecise language usage, such as “Analysis of BcABF1 Gene Expression Patterns, “To obtain the expression pattern”.

5. Typically, figures should be kept at high resolution. It is necessary to improve the sharpness of Figure 4.

6. The experimental findings of Result 5 should be elaborated in detail.

7. In the Materials and Methods section 4.3, a description of the GUS enzyme activity assay method should be supplemented.

8. It is advised to include additional recent references in the manuscript to ensure its relevance and alignment with current research trends.

Comments on the Quality of English Language

 Minor editing of English language required.

Reviewer 3 Report

Comments and Suggestions for Authors

The study addresses an interesting topic regarding the function of the BcABF1 gene coding for a key transcription factor in the ABA signal transduction pathway in Brassica campestris, commonly known as Pakchoi. The authors have elaborated an array of current molecular tools and methods to assess gene expression pattern in various plant tissues and to further our understanding on the interplay with selected promoter regions using a yeast One-Hybrid system. Furthermore, a series of Arabidopsis overexpressing lines were constructed to discern the role of BcABF1 to drought stress as well as the interplay with ABA responsive elements and plant hormones.  The experimental plan and design are well-organized and presented and the study provides a new insight. However, there are a few topics that the authors should elaborate and clarify to improve the manuscript. These are as follows:

1.      In the Introduction ln 56-69 should be edited or rewritten to clearly illustrate a coherent text describing the process in the Arabidopsis model plant. Also reference to genes with their initials should be accompanied by the full name in parenthesis upon first note (i.e. ADF5, IDD14, etc) so to easily understand the interplay.

2.      Ln 86 “characterized by its short growth cycle” it would be nice to indicate the days off the species life cycle.

3.      Ln 97-98, 268, 438 the term “seven cruciferous plants” should be rephrased to “seven cruciferous plant species” to clearly depict the breadth of the plant species used in the analyses of the study.

4.      In section 2.1 “Phylogenetic Analysis of ABF1” the authors should indicate whether there are other ABF homologs in pakchoi, besides the ABF1. Also, this information could be elaborated in the introduction section regarding the homologs of ABF gene found in the Brassica species used in the study, as similar information is provided for the Arabidopsis model.  

5.      In section 2. 2. “Analysis of BcABF1 Gene Expression Patterns” the subtitle should be rephrased omitting the term “patterns”. Gene expression of BcABF1 was determined under dark or light conditions and the sampling took place during the same developmental stage (i.e. days post germination, days-old seedling). These parameters should be clearly indicated. Also is there a difference in tissue specific gene expression in response to light/dark and to the developmental stage of the specific tissue?

6.      Ln 117 “leaf, blot, petiole and flower” what is blot?  Also, the authors are encouraged to carefully check the manuscript for English typo or grammar errors.

7.      In Materials and Methods, ln 406 “Total RNA from plant leaves” should be rephrased to “Total RNA from plant samples” as various plant samples (i.e. roots, leaves, buds, flowers, etc.) were used in the study.

8.      The Figure S2 legend should be elaborated to describe what the promoter regions shown depict, as a vast amount of work has been done, and remains unrecognized.

Comments on the Quality of English Language

Minor english editing.
